# Latent leprosy infection identified by dual RLEP and anti-PGL-I positivity: Implications for new control strategies

**Moises Batista da Silva**[1], **Wei Li**[2], **Raquel Carvalho Bouth**[1], **Angélica Rita Gobbo**[1], **Ana Caroline Cunha Messias**[1], **Tania Mara Pires Moraes**[3], **Erika Vanessa Oliveira Jorge**[1], **Josafá Gonçalves Barreto**[1,4], **Fred Bernardes Filho**[5], **Guilherme Augusto Barros Conde**[6], **Marco Andrey Cipriani Frade**[5], **Claudio Guedes Salgado**[1‡], **John Stewart Spencer**[2‡]*

**1** Laboratório de Dermato-Imunologia, Instituto de Ciências Biológicas, Universidade Federal do Pará, Belem, Pará, Brazil, **2** Colorado State University, Department of Microbiology, Immunology and Pathology, Mycobacteria Research Laboratories, Fort Collins, CO, Unites States of America, **3** Instituto de Saúde Coletiva, Universidade Federal do Oeste do Pará (UFOPA), Santarém, Pará, Brazil, **4** Spatial Epidemiology Laboratory, Universidade Federal do Pará—Campus Castanhal, Castanhal, Pará, Brazil, **5** Division of Dermatology, Department of Internal Medicine of Ribeirão Preto Medical School, University of São Paulo, Ribeirão Preto, São Paulo, Brazil, **6** Laboratório de Suporte a Decisões, Universidade Federal do Oeste do Pará (UFOPA), Santarém, Pará, Brazil

☯ These authors contributed equally to this work.
‡ CGS and JSS also contributed equally to this work.
* john.spencer@colostate.edu

**Data Availability Statement:** All relevant data are within the paper and its Supporting Information files.

## Abstract

The number of new cases of leprosy reported worldwide has remained essentially unchanged for the last decade despite continued global use of free multidrug therapy (MDT) provided to any diagnosed leprosy patient. In order to more effectively interrupt the chain of transmission, new strategies will be required to detect those with latent disease who contribute to furthering transmission. To improve the ability to diagnose leprosy earlier in asymptomatic infected individuals, we examined the combined use of two well-known biomarkers of *M. leprae* infection, namely the presence of *M. leprae* DNA by PCR from earlobe slit skin smears (SSS) and positive antibody titers to the *M. leprae*-specific antigen, Phenolic Glycolipid I (anti-PGL-I) from leprosy patients and household contacts living in seven hyperendemic cities in the northern state of Pará, Brazilian Amazon. Combining both tests increased sensitivity, specificity and accuracy over either test alone. A total of 466 individuals were evaluated, including 87 newly diagnosed leprosy patients, 52 post-treated patients, 296 household contacts and 31 healthy endemic controls. The highest frequency of double positives (PGL-I+/RLEP+) were detected in the new case group (40/87, 46%) with lower numbers for treated (12/52, 23.1%), household contacts (46/296, 15.5%) and healthy endemic controls (0/31, 0%). The frequencies in these groups were reversed for double negatives (PGL-I-/RLEP-) for new cases (6/87, 6.9%), treated leprosy cases (15/52, 28.8%) and the highest in household contacts (108/296, 36.5%) and healthy endemic controls (24/31, 77.4%). The data strongly suggest that household contacts that are double positive have latent disease, are likely contributing to shedding and transmission of disease to their close contacts and are at the highest risk of progressing to clinical disease. Proposed

**Funding:** This work was supported by CNPq (486183/2013-0 CNPq grant for MBS; 448741/2014-8 grant for JGB and 428964/2016-8 grant and 313633/2018-5 scholarship for CGS); CAPES (BEX 6907/14-8 scholarship for MBS, and 157512-0 scholarship for JGB; CAPES PROAMAZONIA 3288/2013; Fulbright Scholar to Brazil 2015-2016 and 2019-2020 (JSS); and The Heiser Program of the New York Community Trust for Research in Leprosy (JGB, MBS, CGS and JSS) grants P15-000827, P16-000796 and P18-000250. The funders had no role in study design, data collection and analysis, decision to publish, or preparation of the manuscript.

**Competing interests:** The authors have declared that no competing interests exist.

strategies to reduce leprosy transmission in highly endemic areas may include chemoprophylactic treatment of this group of individuals to stop the spread of bacilli to eventually lower new case detection rates in these areas.

## Introduction

Leprosy, caused by the human pathogen *Mycobacterium leprae* (*M. leprae*), causes a slowly developing granulomatous disease that affects mainly skin and peripheral nerves, resulting in disfiguring skin lesions and progressive nerve damage with subsequent muscle weakness, bone and tissue resorption, with disfigurement and disability causing stigma and social isolation. Over 200,000 new cases of leprosy have been diagnosed annually in the world during the last 10 years. It is still a public health problem in some endemic areas, particularly in India, Brazil and Indonesia, where 79% of all cases were reported in 2019 [1]. Moreover, independent data allow us to conclude that the real prevalence is much higher than those reported numbers [2, 3]. The global hidden prevalence is estimated at 3 million cases, but can be six times higher than the registered prevalence in some areas [4]. Recent mathematical models predict that elimination of leprosy as a public health risk by 2061 would require over 40 years in the three regions with the highest prevalence in Brazil (North, Northeast and Midwest), primarily due to the long delay in detection of cases [5]. Currently, Brazil is still the only country in the world that has not met the WHO goal of <1 new case per 10,000 population, with 27,863 new cases detected in 2019, around 1.3/10,000 nationally based on Brazil's National Notifiable Diseases Information System (SINAN) [6].

There is no laboratory test capable of detecting all clinical forms of leprosy. Thus, the diagnosis is essentially based on clinical examination of skin and peripheral nerves, ideally by trained dermatologists or leprosy clinicians. However, diagnosing leprosy in Brazil is often made by health care personnel in the basic health units, who may lack training in diagnosing leprosy. The knowledge and skills of leprosy diagnosis, treatment and management by general health workers are unsatisfactory, resulting in delayed detection, leading to an increase in physical disabilities, socioeconomic impairment and continued *M. leprae* transmission [7, 8]. Therefore, the development of a more sensitive diagnostic test suitable for early-stage leprosy and for paucibacillary or asymptomatic disease is considered a research priority [9, 10].

The diagnosis of leprosy is based mainly on physical examination to detect clinical signs and symptoms (hypopigmented or scaly skin lesions with loss of sensation; pain or swelling of nerves; weakness of muscles or loss of function). The five Ridley-Jopling forms used to categorize the disease spectrum are polar tuberculoid (TT), borderline tuberculoid (BT), borderline borderline (BB), borderline lepromatous (BL) and polar lepromatous (LL), with an increasing bacterial load in the lepromatous forms [11]. Indeterminate leprosy is an early stage of the disease with ill-defined skin lesions while pure neural leprosy (PNL) occurs when nerves are enlarged without any detectable skin lesions. The form of the disease is used for classifying patients into paucibacillary (PB) or multibacillary (MB) categories that determines the length of treatment with multidrug therapy (MDT) for 6 or 12 months, respectively. In hyperendemic areas in Pará, Brazil, around 70% of individuals diagnosed are classified as MB. Various tests have been developed to assess anti-PGL-I antibody positivity, a known biomarker of *M. leprae* infection, including the standard ELISA assay [12] and lateral flow rapid tests that incorporate synthetic PGL-I (ND-O-BSA) or novel protein glycoconjugates, like NDO-LID [13, 14]. There is an excellent correlation between the bacillary load (BI) and the anti-PGL-I titer, showing a

progressive increase in the titer in lepromatous forms (BB, BL and LL) while the antibody titer is low to negative in tuberculoid forms (TT, BT). The molecular detection of *M. leprae* DNA in earlobe slit skin smears (SSS) or blood [15], skin lesions, nasal swabs or biopsies using standard PCR [16] or quantitative PCR (qPCR) [17, 18] has also been found to be very useful to detect asymptomatic carriers or to diagnose difficult cases. These confirmatory tests are currently being used to aid in the diagnosis of leprosy patients and are among the strategies that have been recommended for implementation for better leprosy control and patient management [19].

In the current cross-sectional study, we have combined the use of the standard ELISA assay to measure the anti-PGL-I titer in serum with the detection of *M. leprae* DNA by PCR amplification of the *M. leprae*-specific repetitive sequence, RLEP, in earlobe SSS samples in a cross-section of leprosy patients, healthy household contacts and healthy endemic controls from seven hyperendemic municipalities in different regions in the northern state of Pará in the Brazilian Amazon.

## Methods

### Ethics statement

This study conforms to the Declaration of Helsinki and the research protocols were approved by the institutional review boards at the Federal University of Pará (UFPA) (IRB protocol CAAE 26765414.0.0000.0018) and Colorado State University (IRB protocols 15-6340H, 18-8369H and 20-8369H). All individuals who agreed to participate read and signed a written informed consent document. In the case of minors, consent was obtained from a parent or guardian of the child. All data were anonymized.

### Study area

Pará state is in northern Brazil, occupying an area of 1,253,164 km$^2$, being the second largest state in Brazil. To include a broad representation of individuals from cities of various sizes, we selected seven municipalities that came from the six mesoregions in this state. The Lower Amazonas region was represented by samples from the city of Santarém (302,667 inhabitants). In the southwest, samples were collected from Senador José Porfírio (11,839 inhabitants). The town of Breves (101,891 inhabitants) represents the Marajó mesoregion. The southeast was represented by the city of Redenção (83,997 inhabitants), while the northeast was represented by the city of Acará (55,513 inhabitants). The metropolitan region of the capital of Belém is represented by the district of Mosqueiro (approximately 27,000 inhabitants), Castanhal (198,294 inhabitants) and the capital city itself (2.3M inhabitants).

### Sampling design and methods

Leprosy is a compulsory notifiable disease in Brazil and patients detected through either clinic-based passive diagnosis or active surveillance have their clinical data and addresses registered in the national notifiable diseases information system (SINAN). At each site visited a random sampling of subjects from the seven cities surveyed was performed using available data for locating treated patients. We also relied on the local community health agents working with the basic healthcare units to assist us in locating households where new cases of leprosy were suspected. These households were visited by our team where new leprosy patients were diagnosed and their household contacts were assessed. All individuals received a free dermatologic exam performed by experienced leprosy clinicians, and the sample of blood and SSS was taken from each person by a trained phlebotomist. Blood was processed by centrifugation at

each site to yield serum that was frozen and transported to the laboratory for analysis by ELISA. Earlobe SSS samples were placed in microcentrifuge tubes containing 70% ethanol and transported to the laboratory for DNA extraction and analysis by PCR. The diagnosis of leprosy by experienced leprosy clinicians was performed using internationally accepted clinical criteria based on the presence of skin lesions with sensory loss and/or nerve damage associated with nerve swelling and pain, muscle weakness or disability. Individuals diagnosed with leprosy received free MDT treatment from their local basic health unit. A total of 466 individuals from different cities in Pará agreed to provide blood and earlobe SSS for the purpose of assessing the anti-PGL-I titer and *M. leprae* DNA positivity by PCR, respectively. The individuals surveyed included 87 newly diagnosed leprosy patients, 52 former patients who had completed their MDT treatment, 296 household contacts (individuals living in a household with at least one confirmed diagnosed case of leprosy) and 31 healthy endemic control (HEC) subjects with no known exposure to a leprosy patient, mainly undergraduate and graduate students attending the Federal University of Pará, Belém, Pará.

## Assessment of anti-PGL-I titer by ELISA

An indirect ELISA was used to measure the anti-PGL-I IgM titer of each of the serum samples tested at a 1:300 dilution using a protocol previously reported [20]. The cut-off for positivity was established at an optical density (O.D.) of 0.295 based on the average plus three times the standard deviation of healthy subjects from a hyperendemic area as reported. The O.D. for each well was read at 490 nm using an ELISA plate reader.

## DNA extraction and RLEP amplification

Total DNA extraction of earlobe SSS samples using the Qiagen Blood & Tissue DNA kit (Qiagen, Germantown, MD) was performed according to the manufacturers' protocol with minor modifications. Amplification of the *M. leprae* repetitive RLEP sequence (up to 37 copies are found within the genome) was achieved using Qiagen Multiplex PCR Master Mix (Qiagen) according to a previously published protocol [21] using the primer pairs LP1 (5'-TGCATGT-CATGGCCTTGAGG -3') and LP2 (5'-CACCGATACCAGCGGCAGAA-3') described to amplify a 129-base pair fragment found in the genome [22].

## Statistical analysis

The Mann-Whitney U test was used to compare the titers of anti-PGL-I IgM between groups, Student's t-test (unpaired and nonparametric) to evaluate IgM anti-PGL-I titers, and Fisher's exact test was used to compare the proportion of new cases detected among seropositive and seronegative individuals and to calculate the correlation between anti-PGL-I IgM and RLEP amplification. Sensitivity, accuracy, and specificity were determined by exact method of Clopper and Pearson. All analyses were performed using GraphPad Prism version 6.0.

## Results

A total of 466 individuals chosen from seven different municipalities were divided into four groups: 87 newly diagnosed leprosy patients, 52 treated patients who had completed MDT (averaging 4 years after completing treatment), 296 HHC who did not have any clinical signs of disease and 31 HEC (Table 1). Children under the age of 15 years old were included. Of the total number evaluated, 92/466 (19.7%) were children; 38/87 (43.7%) in the new case group; 3/52 (5.8%) in the treated group; 51/296 (17.2%) in the HHC group; and 0/31 (0%) in the HEC group.

**Table 1. Operational classification and number of subjects per group and municipality.** Characteristics of newly diagnosed leprosy patients, treated leprosy patients, healthy household contacts (HHC) and healthy endemic controls (HEC) from the seven cities surveyed.

| City | New leprosy cases | | | | Treated leprosy patients | | | | HHC | HEC |
|---|---|---|---|---|---|---|---|---|---|---|
| | PB | % | MB | % | PB | % | MB | % | | |
| Acará | 4 | 50.0 | 4 | 50.0 | 1 | 10.0 | 9 | 90.0 | 66 | - |
| Breves | 1 | 20.0 | 4 | 80.0 | 1 | 100.0 | - | - | 18 | - |
| Castanhal | - | - | - | - | 1 | 20.0 | 4 | 80.0 | 26 | - |
| Belém/Mosqueiro | 6 | 14.6 | 35 | 85.4 | - | - | - | - | 74 | 31 |
| Redenção | 1 | 25.0 | 3 | 75.0 | 2 | 20.0 | 8 | 80.0 | 24 | - |
| Santarém | 4 | 16.7 | 20 | 83.3 | 6 | 37.5 | 10 | 62.5 | 52 | - |
| Senador José Porfirio | - | - | 5 | 100.0 | 3 | 30.0 | 7 | 70.0 | 36 | - |
| Total | 16 | 18.4 | 71 | 81.6 | 14 | 26.9 | 38 | 73.1 | 296 | 31 |

We first examined the anti-PGL-I titer in serum samples from all individuals. As shown in **Fig 1A** the percent positivity was 55.2% for new cases, 50% in treated individuals, 51.7% in HHC and 22.6% in HEC. There was no statistical difference in the overall percentage of anti-PGL-I positivity between the patient and HHC groups, while positivity in HEC was significantly lower than all other groups. The O.D. for each individual was plotted for the three groups to determine the range and the median O.D. for each group (**Fig 1B**). Although the median O.D. was somewhat higher in the treated group, it was not significantly higher than the median O.D. for new cases or household contacts. The median for the HEC group was the lowest, with only seven individuals slightly exceeding the cut-off. Within the new case group, detection of a positive anti-PGL-I titer was 68.8% (42/61) in MB cases and 37.5% (6/16) in PB cases. When cases were subdivided according to Ridley-Jopling classification for the different forms across the disease spectrum, anti-PGL-I was positive in 57% (4/7) for the indeterminate form, 20% (1/5) for TT, 70.8% (34/48) for BT, 31.6% (6/19) for BB and 50% for BL and LL (2/4), respectively. In addition, four cases of primary neural form (PNL) were diagnosed, and 25% (1/4) were positive.

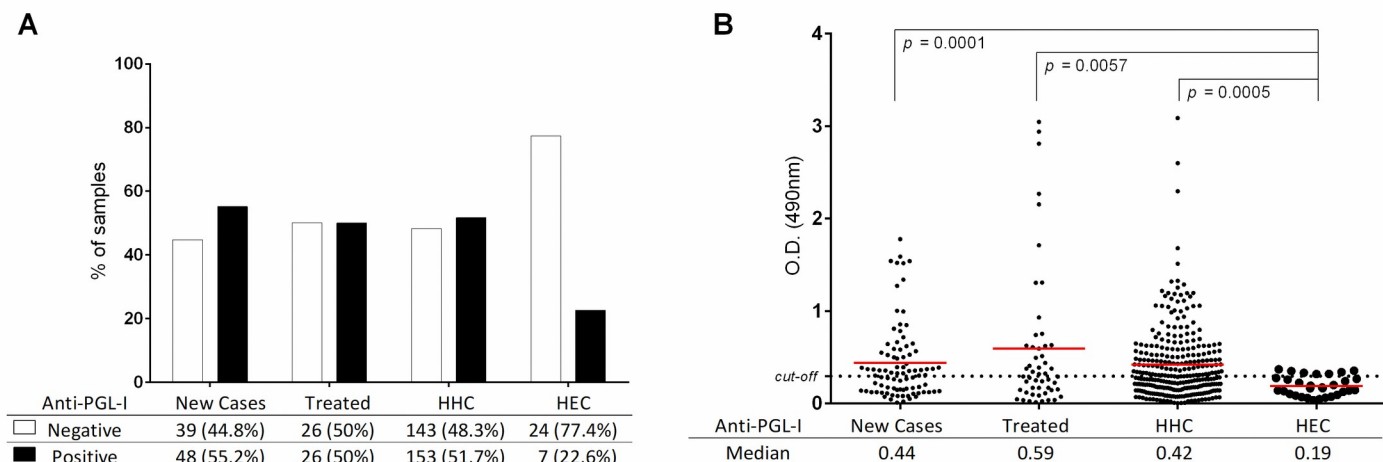

**Fig 1. Frequency of anti-PGL-I positivity in new cases, treated cases, HHC and HEC.** A) Positivity versus negativity in anti-PGL-I titers with a similar percentage of positives observed in newly diagnosed cases, treated cases, and household contacts, while those in the HEC group were negative or weakly above the cut-off. B) Anti-PGL-I optical density (O.D.) for all individuals was plotted for each group with the median O.D. indicated by the solid horizontal line. The significant *p* value differences between groups are shown.

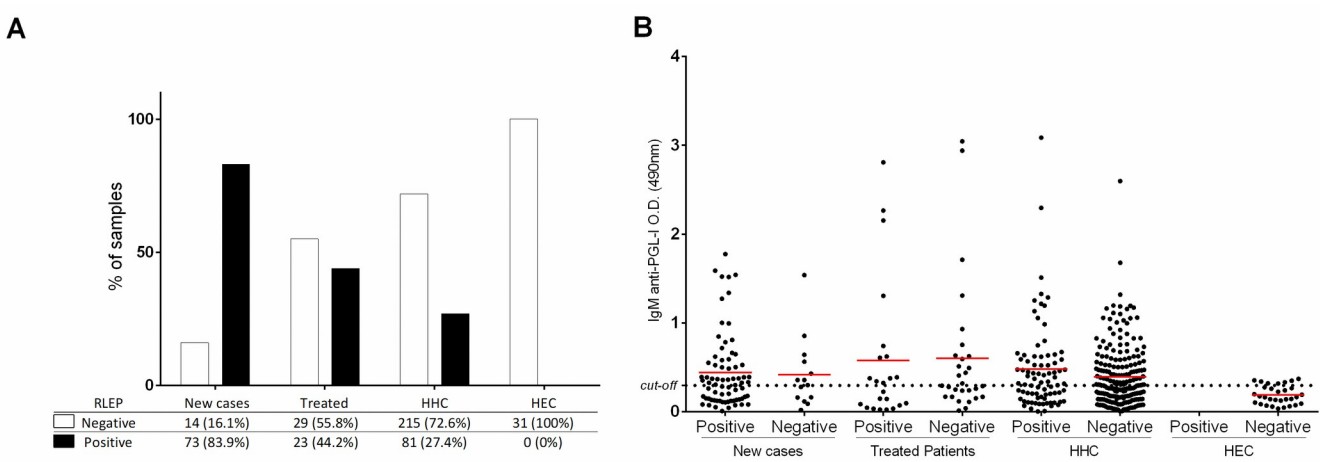

**Fig 2. Analysis of RLEP and PGL-I titer for new cases, treated cases, HHC and HEC.** A) RLEP positivity within each of the four groups examined based on PCR of earlobe SSS. New cases were 83.9% positive, treated cases were 44.2% positive, HHC were 27.4% positive whereas no HEC were positive (0%). B) anti-PGL-I titer was plotted for each individual based on being RLEP positive or negative within each group. Solid line indicates the median O.D. for each group. There was no significant difference between the median anti-PGL-I titer when positive and negative RLEP groups were compared between any of the two patient and HHC groups.

RLEP amplification was performed on DNA prepared from earlobe SSS to assess the presence of *M. leprae* DNA (**Fig 2A**). In the new case group, 73/87 individuals (83.9%) were RLEP positive; in the treated group (median 4 years after finishing treatment), 23/52 (44.2%) were positive; in the HHC group, 81/296 (27.4%) were positive, while in the HEC group, none were positive (0/31). When individuals in each group were divided into RLEP positive or negative and examined for anti-PGL-I titer, there were no statistical differences between the median O. D. values for RLEP positive versus negative individuals within the patient or HHC groups (**Fig 2B**). Within the new case group, detection of RLEP was 87.3% (62/71) in MB cases and 68.8% (11/16) in PB cases. When cases were subdivided according to Ridley-Jopling classification for the different forms across the disease spectrum, RLEP amplification was positive in 57.1% (3/7) for the indeterminate form, 80% (4/5) for TT, 87.5% (42/48) for BT, 84.2% (16/19) for BB and 100% for BL and LL (4/4). In addition, four cases of primary neural form (PNL) were diagnosed, and all were positive (100%, 4/4).

Finally, we examined both RLEP and anti-PGL-I results within the four groups of subjects. The results showed the highest frequency of double positives (PGL-I+/RLEP+) in the new case group (40/87, 46%) with lower numbers for treated (12/52, 23.1%), HHC (46/296, 15.5%) and none for HEC (0/31, 0%). The frequencies in the groups were reversed for double negatives (PGL-I-/RLEP-) for new cases (6/87, 6.9%), treated cases (15/52, 28.8%) and the highest in HHC (108/296, 36.5%) and HEC (24/31, 77.4%) (**Table 2**). We did not detect any differences in sensitivity of anti-PGL-I or RLEP PCR detection based on either the geographic origin of the samples, nor were there differences based on whether the samples were from male or female subjects.

IgM anti-PGL-I serology showed 55% sensitivity, 51% specificity and 52% accuracy, while detection of RLEP DNA in SSS was higher, with sensitivity of 84%, specificity of 75% and accuracy of 77%. Combining both laboratory tests increased sensitivity to 88%, specificity to 77%, and accuracy to 79%.

The socioeconomic data collected from all of the individuals studied shown in Table 3 revealed that more than 20% of new cases and 34% of treated cases and HHC declared they had suffered food deprivation at least once during the year, while food insecurity did not exist

**Table 2. Correlation of RLEP and anti-PGL-I titer within each group.**

| | PGL +/RLEP+ | | PB | | MB | | PGL-/RLEP+ | | PB | | MB | | PGL +/RLEP- | | PB | | MB | | PGL-/RLEP- | | PB | | MB | |
|---|---|---|---|---|---|---|---|---|---|---|---|---|---|---|---|---|---|---|---|---|---|---|---|---|
| | n | % | n | % | n | % | n | % | n | % | n | % | n | % | n | % | n | % | n | % | n | % | n | % | n | % |
| New cases (n = 87) | 40 | 46.0 | 4 | 10 | 36 | 90 | 33 | 37.9 | 7 | 21.2 | 26 | 78.8 | 8 | 9.2 | 2 | 25 | 6 | 75 | 6 | 6.9 | 3 | 50 | 3 | 50 |
| Treated (n = 52) | 12 | 23.1 | 4 | 33.3 | 8 | 66.7 | 11 | 21.2 | 4 | 36.4 | 7 | 63.6 | 14 | 26.9 | 2 | 14.3 | 12 | 85.7 | 15 | 28.8 | 5 | 33.3 | 10 | 66.7 |
| HHC (n = 296) | 46 | 15.5 | | | | | 35 | 11.8 | | | | | 107 | 36.1 | | | | | 108 | 36.5 | | | | |
| HEC (n = 31) | 0 | 0% | | | | | 0 | 0% | | | | | 7 | 22.6 | | | | | 24 | 77.4 | | | | |

Double positive (PGL-I+/RLEP+), single positive (PGL-I+/RLEP- or PGL-I-/RLEP+) and double negative (PGL-I-/RLEP-) were calculated for each of the four groups. The numbers of PB and MB cases are shown for the new case and treated case groups.

in the HEC group. The financial situation of many of the families was quite precarious, showing that 63.2% of new cases, 57.7% of treated cases and 58.1% of HHC had ≤ one Brazilian minimum salary per month (approximately $200 USD), while none of the HEC fell into this category. A source of good clean drinking water (filtered or mineral water versus untreated, strained or chlorinated water) was lacking for 51.7% of new cases, 59.6% of treated cases, and 65.5% of HHC, while none of the HEC were without clean drinking water. Low (only elementary schooling) or no education was recorded among 88.5% of new cases, 73.1% of treated cases and 78.7% of HHC, while 100% of the HEC were university educated. Finally, there was a much higher density in living conditions in households of new cases, treated cases and HHC averaging between 5.0 to 5.5 individuals, while HEC households had on average only 3 individuals living together.

## Discussion

Bacilloscopy is the gold standard laboratory test to detect acid fast *M. leprae* in the skin of diagnosed leprosy patients, and is important to establish the bacillary index (BI, a logarithmic scale of the number of acid fast bacilli detected in the skin, where 0 is none detected and 6+ is the highest) for determining the treatment regimen (MB or PB, one year or 6 months MDT treatment, respectively). However, this test, although highly specific, has a low sensitivity (only 44%) [23], is labor intensive, requiring experienced lab personnel and may require taking a skin punch biopsy, which is somewhat invasive. Thus, it is performed only in presumed leprosy cases, and is negative in the majority of PB or primary neural forms. Biopsies are not performed on household contacts since, absent any lesions, almost all would be negative. For this reason, we have been using less invasive methods, namely taking samples of blood and earlobe SSS to assess anti-PGL-I positivity by ELISA and the presence of *M. leprae* DNA by PCR, respectively. Each of these biomarkers of infection pose an independent risk for an individual to progress to disease. People who are positive for anti-PGL-I have about a 6-fold higher risk of progressing to disease [24], and we have previously established that by following 10 seropositive individuals, there is a >90% chance that one of these individuals will progress to disease within a two-year timeframe [25]. Similarly, confirming molecular amplification of *M. leprae* DNA in nasal or oral mucosa [26, 27] or skin biopsies or smears can be used as a biomarker of infection, indicating colonization in the skin. After infection of the nasal mucosa by *M. leprae*, one of the secondary sites of infection is the earlobe due to its proximity to the nose and is a preferred site because of its relative coolness. Demonstration of *M. leprae* infection of the earlobe by identifying acid fast bacilli in SSS or by detecting a positive PCR for bacterial DNA can be used as confirmatory evidence in the diagnosis of a patient with other clinical signs of disease. Studies have shown that the use of the repetitive element RLEP as the target in detecting

**Table 3. Demographic information of the four groups studied (new leprosy cases, treated patients, HHC and HEC) including household density (average number of people living in the house), type of water used for drinking and cooking, income level, education level, receiving governmental support, median age and range, ratio of number of males to females, incidence of food deprivation and living in an urban versus a rural area.**

| | Number of people per house | Type of water used for drinking/cooking | Salary | Education (highest grade) | Receive governmental support (%) | Median age (range) | Sex (ratio M:F) | Food deprivation (%) | Living in urban area (%) |
|---|---|---|---|---|---|---|---|---|---|
| New cases (n = 87) | 5.5 | Not treated: 4 Strained water: 39 Chlorinated: 2 Filtered: 36 Mineral water: 6 | ≤ 1 minimum salary: 55 Up to two minimum salary: 21 Up to three minimum salary: 5 Greater than 3MS: 6 | No Education: 66 Elementary School: 11 High school: 9 University education: 1 | 60/87 (69%) | 25 (5–81) | Female: 52 Male: 35 (0.6: 1) | 18/87 (20.7%) | 81/87 (93.1) |
| Treated (n = 52) | 5 | Not treated: 3 Strained water: 21 Chlorinated: 7 Filtered: 12 Mineral water: 5 | ≤ 1 minimum salary: 30 Up to two minimum salary: 13 Up to three minimum salary: 4 Greater than 3MS: 3 | No Education: 30 Elementary School: 8 High school: 8 University education: 6 | 35/52 (67.3%) | 45 (12–87) | Female: 22 Male: 30 (1: 0.58) | 18/52 (34.6%) | 42/52 (80.8%) |
| HHC (n = 296) | 5 | Not treated: 15 Strained water: 147 Chlorinated: 32 Filtered: 70 Mineral water: 32 | ≤ 1 minimum salary: 172 Up to two minimum salary: 81 Up to three minimum salary: 27 Greater than 3MS: 16 | No Education: 106 Elementary School: 127 High school: 52 University education: 11 | 193/296 (65.2%) | 31 (6–79) | Female: 160 Male: 136 (0.46: 1) | 102/296 (34.4%) | 241/296 (81.4%) |
| HEC (n = 31) | 3 | Not treated: 0 Strained water: 0 Chlorinated: 0 Filtered: 2 Mineral water: 29 | ≤ 1 minimum salary: 0 Up to two minimum salary: 1 Up to three minimum salary: 2 Greater than 3MS: 28 | No Education: 0 Elementary School: 0 High school: 0 University education: 31 | 0/31 (0%) | 33 (19–62) | Female: 20 Male: 11 (0.35: 1) | 0/31 (0%) | 31/31 (100%) |

*M. leprae* DNA is more sensitive than single copy genes such as *rpoT*, *sodA* and *16S rRNA* [28, 29], so for this reason we used primers to detect the RLEP sequence.

Although it has been previously established that being anti-PGL-I positive puts an individual at higher risk for eventually coming down with leprosy [30–32], our previous studies in cities in the state of Pará have shown that seropositivity among residents living in hyperendemic areas is generally quite high, usually 40–60%, reflecting the very high levels of *M. leprae* circulating in the general population [2, 20, 21, 25, 33, 34]. Despite these high rates of positivity, there is an overall genetic resistance towards developing leprosy, with over 90% of people having a natural immunity [35]. Even in this study, anti-PGL-I positivity in newly diagnosed leprosy patients was only slightly higher than in HHC, 55.2% versus 51.7%, respectively. The main reason for this is that the majority of newly diagnosed leprosy patients were classified as

BT (48/87, 55.2%), known for having low or no anti-PGL-I antibody titer. For this reason, we wanted to pair an additional biomarker of *M. leprae* infection, namely RLEP PCR of earlobe SSS, to determine if the two biomarkers together could be more informative as far as identifying those with latent disease. For example, we discovered that some of the households we examined in the town of Acará, just two hours from the capital of Belém, had extremely high percentages of anti-PGL-I and RLEP PCR positivity. In one household with 12 people living together with a newly diagnosed index case, 92% (11/12) were found to be anti-PGL-I positive, 75% (9/12) had evidence of *M. leprae* colonization of their earlobes by RLEP PCR, and 75% were positive for both biomarkers. Only one individual from this family was negative for both biomarkers (anti-PGL-I-/RLEP-). At the time of visiting this household, six other blood related family members were clinically diagnosed, supporting published reports that HHC who have a genetic relatedness with an untreated MB index case have the highest risk of progressing to disease [36, 37]. These very high percentages for anti-PGL-I and RLEP PCR double positivity were seen in several other large extended households in the same neighborhood, indicating high rates of transmission and infection in this area.

These high levels of double positives in large families in many households in hyperendemic settings led us to conduct this current survey in multiple cities to determine if these results were generalizable in different areas of Pará state. Although there were no differences between the three groups (new patients, treated patients and HHC) as far as the percent of anti-PGL-I positivity, there were large differences in the rates of RLEP positivity in these groups. Newly diagnosed cases were overwhelmingly positive by PCR of SSS (83.9%), indicating earlobe colonization, while just under half of individuals who were treated were positive (44.2%) indicating a large reduction in the bacterial burden following treatment in this group. HHC showed the lowest rates of RLEP PCR positivity at only 27.4% even though slightly more than half of this group were anti-PGL-I positive (51.7%). The lack of RLEP positivity in HEC can be explained by the fact that most of these individuals (almost all are university students) come from a higher socioeconomic background and have no known contact with a leprosy patient. When these two biomarkers of infection were paired together, it revealed more compelling information. Almost half of newly diagnosed patients were double positive (anti-PGL-I+/RLEP PCR+, 46%), while a minority in this group were double negative (anti-PGL-I-/RLEP PCR-, 6.9%). Double negatives were higher in the treated group (28.8%), likely indicating treatment efficacy. In contrast, HHC had slightly more double negatives overall (36.5%) and much fewer double positives than either of the patient groups (15.5%). Despite this lower number, HHC who are double positive were found to have a much higher risk of progressing towards disease (OR = 19) since they most closely resemble the high rate of double positives found in newly diagnosed patients. For this reason, longitudinal long-term follow-up of these individuals would be critical to understanding their proclivity to progress towards disease over those who are double negative.

The four different possible combinations of ELISA/PCR results can be cautiously interpreted in several ways. We propose that those individuals without clinical signs and symptoms of leprosy who are PGL-I+/RLEP+ have latent leprosy infection, allowing permissive growth to allow infection of *M. leprae* in the earlobe and spread to other sites in the skin and induce an antibody response. These individuals most resemble newly diagnosed patients, the majority of whom are double positive, and thus are at the greatest risk of progressing to disease and spreading it to others. Individuals who are PGL-I+/RLEP- are infected but their functional cell mediated immune response has limited bacterial infection in the earlobe, which can evolve to a cure or can progress to paucibacillary disease. PGL-I-/RLEP+ individuals are also infected but the bacillary load has not increased to the point that induces an anti-PGL-I response. These individuals could either control the bacilli or progress to disease if the cell mediated

response allows permissive growth and spread. Individuals who are double negative, PGL-I-/ RLEP-, may not have been exposed to enough of a bacterial load to infect them or were more resistant to infection. These results could change over time depending on continued exposure to an untreated index case or other factors that can degrade a robust cell mediated immune response (co-infections, poor nutritional status).

Our demographic data confirm the close relationship between leprosy and several socioeconomic indicators. Human development index (HDI) scores that take into account three main metrics including life expectancy at birth, level of education and per capita income show a good correlation between lower HDI and higher new case detection rates throughout Brazil. Meta-analysis of secondary data has shown that poor socioeconomic conditions are associated with an increased risk of acquiring leprosy, with food deprivation and low socioeconomic levels being the most critical [38–41]. Food deprivation has also been indicated as an important factor for the development of clinical leprosy as the insufficient intake of macronutrients/ micronutrients impairs the immune system and decreases host protection, leading to increased frequency and severity of infections [42, 43]. Related to this is the poor quality of drinking water that can be contaminated by disease causing bacteria, amoeba and helminths, co-infections that can cause a shift to a Th2 cytokine profile that does not protect against intracellular infections like *M. leprae*. Amoebal cysts can also carry *M. leprae* that can be ingested by drinking contaminated water [44]. The deficiencies observed in the patient and household contact groups with regard to low income levels, reliance on government aid programs, low educational levels, food insecurity, lack of clean drinking water, overcrowding conditions, lack of primary health care coverage and living in a leprosy hyperendemic environment all contribute to a higher risk of succumbing to leprosy.

Over the last few years, the principal stakeholders, including the WHO, involved in promulgating strategies aimed at reducing the global burden of leprosy, particularly in hot spots or high to hyperendemic regions, agree that early diagnosis, contact tracing, and treatment of all patients should be part of the overall strategy. The reported number of new leprosy cases worldwide has been above 200,000 for the last five years, with cases in children averaging around 8%, indicating continued leprosy transmission, and rates of grade 2 disability hovering around 6%, indicating serious delays in diagnosis [45]. However, there are many poor areas of the world that do not report statistics on leprosy [46] and low levels of contact tracing and follow-up could theoretically lead to large numbers of unreported cases in the coming years [47]. It has been suggested that chemoprophylaxis involving the use of single dose rifampicin (SDR) treatment of contacts of leprosy cases might be one way to reduce the number of cases in high endemic areas. In recent developments, there are large-scale clinical trials underway coordinated by national programs that are examining the efficacy of the use of Leprosy Post-Exposure Prophylaxis (LPEP), either SDR or other multidrug short course therapy regimens in multi-country locations to evaluate the potential of accelerating the reduction of transmission in high and hyperendemic areas [48]. Our results indicate that up to 25% of the contacts in highly endemic areas are already infected by *M. leprae* with no clinical disease. This may be one of the reasons why SDR has not be effective to control leprosy in hyperendemic areas. Instead of SDR, the use of MDT or other short course drug combinations currently being tested may be necessary to treat those infected contacts with latent infectious disease [49].

Considering all of the available data, it might be possible to target HHC that are double positive (anti-PGL-I+/RLEP PCR+), as these individuals have latent leprosy infection, are probably shedding bacilli and contributing to infecting their household contacts and are therefore most likely to progress to disease. Combining both of these tests increased the sensitivity and specificity over either test alone and may provide added benefit to detecting those with latent leprosy. Prophylactic treatment of this high-risk group and their HHC would likely be an

effective strategy to end transmission within all of the contacts of these households. There are strong hopes that the use of these kinds of aggressive strategies will ultimately break the lines of transmission and successfully remove leprosy as a major health concern.

## Limitations of the study

The diagnosis of leprosy is still based on the detection of classic signs and symptoms of skin lesions with loss of sensation, nerve damage with loss of function, swelling or pain and visible deformity as detected by well-trained clinicians or health care personnel since *M. leprae* cannot be cultivated. Despite the use of adjunct laboratory tests to detect acid fast bacilli in skin smears or *M. leprae* DNA by PCR, these tests are not always available in resource constrained settings. The anti-PGL-I assay, although easy to perform, only indicates prior infection by the bacillus and a positive test by itself does not trigger the administration of MDT since the majority of positive individuals will not progress to clinical disease. Although we have shown that the majority of newly diagnosed leprosy cases with clinical symptoms are positive for both anti-PGL-I and *M. leprae* DNA by PCR, whether household contacts who are also double positive have latent disease and are at the highest risk of succumbing to disease can only be determined in future long term longitudinal follow-up studies.

## Supporting information

**S1 Fig. Frequency of anti-PGL-I positivity in new cases, post-treated cases, HHC and HEC.** A) Positivity versus negativity in anti-PGL-I titers with a similar percentage of positives observed in newly diagnosed cases, post-treated cases, and household contacts, while those in the HEC group were negative or weakly above the cut-off. B) Anti-PGL-I optical density (O.D.) for all individuals were plotted for each group with the median O.D. indicated by the solid horizontal line. The significant *p* value differences between groups are shown.
(DOC)

**S2 Fig. Analysis of RLEP and PGL-I titer for new cases, post-treated cases, HHC and HEC.** A) RLEP positivity within each of the four groups examined based on PCR of earlobe SSS. New cases were 83.9% positive, treated cases were 44.2% positive, HHC were 27.4% positive whereas no HEC were positive (0%). B) anti-PGL-I titer was plotted for each individual based on being RLEP positive or negative within each group. Solid line indicates the median O.D. for each group. There was no significant difference between the median anti-PGL-I titer when positive and negative RLEP groups were compared between any of the two patient and HHC groups.
(DOC)

**S1 Table. Operational classification and number of subjects per group and municipality.** Characteristics of newly diagnosed leprosy patients, treated leprosy patients, healthy household contacts (HHC) and healthy endemic controls (HEC) from the seven cities surveyed.
(DOC)

**S2 Table. Correlation of RLEP and anti-PGL-I titer within each group.** Double positive (PGL-I+/RLEP+), single positive (PGL-I+/RLEP- or PGL-I-/RLEP+) and double negative (PGL-I-/RLEP-) were calculated for each of the four groups. The numbers of PB and MB cases are shown for the new case and treated case groups.
(DOCX)

**S3 Table. Demographic information of the four groups studied (new leprosy cases, treated patients, HHC and HEC) including household density (average number of people living in**

**the house), type of water used for drinking and cooking, income level, education level, receiving governmental support, median age and range, ratio of number of males to females, incidence of food deprivation and living in an urban area.**
(DOCX)

## Author Contributions

**Conceptualization:** Josafá Gonçalves Barreto, Claudio Guedes Salgado.

**Data curation:** Moises Batista da Silva, Wei Li, Raquel Carvalho Bouth, Angélica Rita Gobbo, Tania Mara Pires Moraes, Erika Vanessa Oliveira Jorge, Josafá Gonçalves Barreto, Fred Bernardes Filho, Guilherme Augusto Barros Conde, Marco Andrey Cipriani Frade.

**Formal analysis:** Moises Batista da Silva, Ana Caroline Cunha Messias, Josafá Gonçalves Barreto.

**Funding acquisition:** Claudio Guedes Salgado, John Stewart Spencer.

**Investigation:** Raquel Carvalho Bouth, Angélica Rita Gobbo, Ana Caroline Cunha Messias, Tania Mara Pires Moraes, Erika Vanessa Oliveira Jorge, Josafá Gonçalves Barreto, Fred Bernardes Filho, Guilherme Augusto Barros Conde, Marco Andrey Cipriani Frade, Claudio Guedes Salgado.

**Methodology:** Moises Batista da Silva, Wei Li, Raquel Carvalho Bouth, Angélica Rita Gobbo, Guilherme Augusto Barros Conde, John Stewart Spencer.

**Project administration:** Josafá Gonçalves Barreto, Claudio Guedes Salgado.

**Resources:** Claudio Guedes Salgado, John Stewart Spencer.

**Software:** Guilherme Augusto Barros Conde.

**Supervision:** Moises Batista da Silva, Raquel Carvalho Bouth, Angélica Rita Gobbo, Erika Vanessa Oliveira Jorge, Josafá Gonçalves Barreto, Fred Bernardes Filho, Marco Andrey Cipriani Frade, Claudio Guedes Salgado, John Stewart Spencer.

**Validation:** Moises Batista da Silva, Josafá Gonçalves Barreto.

**Visualization:** Moises Batista da Silva.

**Writing – original draft:** Moises Batista da Silva, Claudio Guedes Salgado, John Stewart Spencer.

**Writing – review & editing:** Moises Batista da Silva, Josafá Gonçalves Barreto, Marco Andrey Cipriani Frade, Claudio Guedes Salgado, John Stewart Spencer.

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
