## [Decision Letter · Decision Letter 0]

11 Sep 2020

PONE-D-20-19814

RLEP and anti-PGL-I double positivity in leprosy patient household contacts represents an important source of transmission in hyperendemic cities in Pará, Brazil.

PLOS ONE

Dear Dr. Spencer,

Thank you for submitting your manuscript to PLOS ONE. After careful consideration, we feel that it has merit but does not fully meet PLOS ONE’s publication criteria as it currently stands. Therefore, we invite you to submit a revised version of the manuscript that addresses the points raised during the review process.

We look forward to receiving your revised manuscript.

Kind regards,

Rashid Ansumana

Academic Editor

PLOS ONE

Journal Requirements:

Reviewers' comments:

Reviewer's Responses to Questions

**Comments to the Author**

1. Is the manuscript technically sound, and do the data support the conclusions?

Reviewer #1: Partly

Reviewer #2: Yes

Reviewer #3: No

Reviewer #4: Yes

Reviewer #5: Yes

2. Has the statistical analysis been performed appropriately and rigorously? 

Reviewer #1: Yes

Reviewer #2: Yes

Reviewer #3: No

Reviewer #4: Yes

Reviewer #5: I Don't Know

3. Have the authors made all data underlying the findings in their manuscript fully available?

Reviewer #1: Yes

Reviewer #2: Yes

Reviewer #3: No

Reviewer #4: Yes

Reviewer #5: Yes

4. Is the manuscript presented in an intelligible fashion and written in standard English?

Reviewer #1: No

Reviewer #2: Yes

Reviewer #3: No

Reviewer #4: Yes

Reviewer #5: Yes

5. Review Comments to the Author

Reviewer #1: Dear authors,

please make your manuscript shorter and focus mainly on your finding

line 412"Prophylactic treatment of this high risk group and their HHC would likely be an effective strategy.." what did you mean from prophylactic treatment? MDT or SDR?

Reviewer #2: The design and research of the paper are highly operable. It has certain value for the diagnosis and prevention of leprosy. The disadvantage is that the sensitivity of the test is low, and the leprosy patients cannot be completely screened.

Reviewer #3: Comments to the authors

The overall recommendation to the authors is they should well acquit themselves with the STROBE guidelines and re-write the paper (I’m assuming the study design is cross-sectional). The paper has an overall message they are trying to communicate but it is not supported by the methods, results and the discussion.

Abstract

The abstract vaguely reports what the problem is, what is known about the problem and what the authors did to solve the problem.

However on introducing the bio-markers they should have ideally been written in full. Then abbreviations could be used thereafter (line 40 and 41).

Introduction

The introduction overall fairy tries to highlight what the problem is, what the gaps are and what was done about it. However:

(Line 118) The Dr. Marcelo Candia Reference Unit in Sanitary Dermatology (UREMC), I don’t understand the relevance of this clinic. Does it offer the standard testing; is it the Center of Excellence in Para? More clarity maybe needed for the reader.

Line 144-Line 149 Are actually Methods and Materials in the introduction section. The study design has been highlighted as well as the study description. I think this should be moved to the methods section.

Line 149-line 153 this is actually the discussion section in the introduction. This part should be moved to the discussion section.

Methods

Overall, the methods are poorly presented and hence the study cannot be reproduced. We don’t know what study design was used. A prospective study design was mooted in the introduction but it is not anywhere else. I advise the authors to look up the STROBE guidelines by Von Elm et al.

This the study is about probable diagnostic tests, the authors should tell us if the tests were done in series or in parallel since the results change depending on what is done.

There are some result tables within the methods section.

Results

I still advise the authors to have a look at the STROBE guidelines. However, it is important to add the socio-demographics of the study subjects unless this is a secondary analysis. The socio-demographics will help us with generalization and also perspective.

The author should use the statistical methods mentioned in the results section to find out if they are significant. Currently all we have are the proportions. We are not sure if they are purely by chance.

Discussion

Overall the discussion is about the bio-markers and leprosy in general. Very little effort is made to discuss the results. Their significance and the recommendations if any, the overall recommendation is hinted all over the paper with no possible justification given from the results.

Reviewer #4: 1) This is a good study trying to address the issue of transmission in leprosy

2) In fact this investigation is trying to look at possible sources of infection in the community and their role in transmission of the disease

3) In Table 2 for all parameters the author should give number of MB and PB cases in each group

4) In HCC it will be interesting to know the type of Index cases whether they are PB or MB

5) If skin smear BI available in Index case was there a correlation between high BI positivity and HCC positive

6) Is there a correlation between smear positivity (If available) and positivity in the two tests performed?

7) In HCC was there any other household which had many members who were positive for any of the two tests and the author should mention about number of families which were positive.

8) In HCC compare the positivity in different age groups and was there any correlation with higher age group?

9) The author needs to mention about the age groups of subjects and among the cases positive for any test in all groups, was there any child case?

10 There is no legend for the Tables mentioned

11) How many HHC who were doubly positive were followed up if followed up and what was the outcome of that follow up?

Reviewer #5: Leprosy is still a public health problem in some countries. In Brazil, the highest prevalence of the disease is observed in North, Northeast and Midwest regions. One important fact regarding the elimination of leprosy is the absence of a gold standard test for diagnosis that results in a significant number of hidden cases. Prophylactic treatment of contacts from multibacillary patients has been evaluated as an effective strategy to control the disease, but the identification of contacts with subclinical infection is important to determine the targets of chemoprophylactic strategy. Here, Silva and colleagues described that contacts that are positive for both RLEP and PGL-1 have latent disease and are at highest risk of progressing to clinical disease. This paper is interesting. However, I have some concerns:

1. Although the experimental design seems correct, authors need to describe the methodology for analyzing the data. How did they calculate sensitivity, specificity and accuracy? Please include in the methodology section.

2. The "Discussion" is rather long and sometimes confusing. It contains common information that is not directly related to the specific topic of this study.

3. Several studies have reported that PGL-1 is a marker of exposure, but not necessarily of infection. It is not clear in the discussion what is the hypothesis for the different profiles observed in the contacts. For example, PGL-1+RLEP+ are the latent patients. But, what is the hypothesis for PGL-1-RLEP+? Please discuss it.

6. PLOS authors have the option to publish the peer review history of their article (what does this mean?). If published, this will include your full peer review and any attached files.

Reviewer #1: **Yes: **Azin Ayatollahi, MD

Reviewer #2: No

Reviewer #3: **Yes: **Michael Kakinda

Reviewer #4: No

Reviewer #5: **Yes: **Roberta Olmo Pinheiro

---

## [Author Response · Author response to Decision Letter 0]

10 Nov 2020

We thank the reviewers very much for their questions and comments which were very helpful. We have endeavored to answer all of the questions and make the changes and additional clarifying paragraphs to the manuscript based on these suggestions wherever possible, keeping in mind that the reviewers also asked us to shorten the manuscript where possible. Our responses are in blue after each reviewer comment. We hope that the revised version of this manuscript is improved from the original. 

PONE-D-20-19814

RLEP and anti-PGL-I double positivity in leprosy patient household contacts represents an important source of transmission in hyperendemic cities in Pará, Brazil.

Revised title: Latent leprosy infection identified by dual RLEP and anti-PGL-I positivity: Implications for new control strategies.

PLOS ONE

Please submit your revised manuscript by Nov 4, 2020 11:59PM. Please upload your review as an attachment if it exceeds 20,000 characters

Reviewer #1: 

Dear authors,

Please make your manuscript shorter and focus mainly on your findings

Thank you for these suggestions, we will endeavor to make changes wherever possible to achieve this.

line 412 "Prophylactic treatment of this high risk group and their HHC would likely be an effective strategy." what did you mean from prophylactic treatment? MDT or SDR?

This is a very important question that we can address in this response, but we will not debate this in our manuscript because it would be too long and is tangential to our study. Studies to examine the efficacy of the broad use of SDR under the LPEP (Leprosy Post Exposure Prophylaxis) trials have been ongoing since 2015, having been implemented alongside national leprosy programs in India, Indonesia, Myanmar, Nepal, Sri Lanka, Tanzania, Brazil and Cambodia. To date, they have not published any data on the efficacy of widespread use of SDR. There has been much debate among leprosy clinicians and academics as to whether SDR is a cost-effective intervention and there are concerns that a single dose of rifampin is not sufficient to protect against developing multibacillary (MB) disease.1 Others are concerned about SDR potentially increasing drug resistance. There may also be ethical problems in telling people that they will be protected since results of the large Chemoprophylaxis of Leprosy (COLEP) trial in Bangladesh found that SDR only protects some people from some types of leprosy (not MB disease) and only for up to 2 years.1 We include a new paragraph in the Discussion to address this. 

1. Lockwood DNJ, Krishnamurthy P, Kumar B, Penna G. 2018. Single-dose rifampicin chemoprophylaxis protects those who need it least and is not a cost-effective intervention. PLoS Negl Trop Dis 12: e0006403. doi.org/10.1371/journal.pntd.0006403

Reviewer #2: 

The design and research of the paper are highly operable. It has certain value for the diagnosis and prevention of leprosy. The disadvantage is that the sensitivity of the test is low, and the leprosy patients cannot be completely screened.

Thank you for these observations. We agree that the sensitivity of the anti-PGL-I ELISA is low in the newly diagnosed patient group (55%), but the majority of these patients are categorized as BT (42/87, 48.3%), and since patients in this category have a low BI, anti-PGL-I responses are generally weak or negative compared to MB patients. Those in the HHC group had a positivity rate slightly less than the new patient group. This is the main reason why we decided to couple anti-PGL-I with RLEP PCR results from earlobe skin smears, as the RLEP positivity in the new patient group was 83.9% (73/87). By detecting the number of double positives in HHC (15.5%) and showing that double positivity represents the largest percentage in newly diagnosed leprosy patients (46%), it suggests to us that this group of HHC has the highest risk of disease progression. Our findings are similar to results in other articles published in the literature, showed a sensitivity of the serological test of 55% in a hyper-endemic population, and 84% for conventional PCR, while the association of both tests reached 88% sensitivity.

The detection sensitivity presented by conventional PCR (88%) to a similar test for tuberculosis approved by the FDA (TB Amplicor-Roche), which presents 79 to 91% sensitivity1, was superior to staining methods, which shows around 59% sensitivity.2 We recognize that greater sensitivity is the goal to be achieved, however other detection tools, such as qPCR and ddPCR, are being tested by us to achieve that goal.

1. Yang S and Rothman RE. 2004. PCR-based diagnostics for infectious diseases: uses, limitations, and future applications in acute-care settings. Lancet Infect Dis 4: 337-48.

2. Girma S, Avanzi C, Bobosha K, Desta K, Idriss MH, Busso P, et al. 2018. Evaluation of Auramine O staining and conventional PCR for leprosy diagnosis: A comparative cross-sectional study from Ethiopia. PLoS Negl Trop Dis 12(9): e0006706. doi.org/10.1371/journal.pntd.0006706

Reviewer #3: 

The overall recommendation to the authors is they should well acquit themselves with the STROBE guidelines and re-write the paper (I’m assuming the study design is cross-sectional). The paper has an overall message they are trying to communicate but it is not supported by the methods, results and the discussion.

Thank you for your suggestions. We will adhere to the STROBE guidelines where possible.

Abstract

The abstract vaguely reports what the problem is, what is known about the problem and what the authors did to solve the problem.

Thank you for your suggestion. Lines 37-45 in the Abstract straightforwardly lay out what the problem is, what is known about the problem and what we did to solve the problem. “In order to more effectively interrupt the chain of transmission, new strategies will be required to detect those with subclinical disease who contribute to spreading disease. To improve the ability to diagnose leprosy earlier in asymptomatic infected individuals, we examined the combined use of two well-known biomarkers of M. leprae infection, namely the presence of M. leprae DNA by PCR from earlobe slit skin smears (SSS) and positive serum antibody to the M. leprae-specific Phenolic Glycolipid I antigen (anti-PGL-I) from leprosy patients and household contacts living in seven hyperendemic cities in the northern state of Para, Brazilian Amazon.”

However on introducing the bio-markers they should have ideally been written in full. Then abbreviations could be used thereafter (line 40 and 41).

Thank you for your suggestion. The term for earlobe slit skin smears (SSS) is defined in line 40, while the presence of testing M. leprae DNA by PCR, these abbreviations should not have to be further explained. The term for the antibody to the M. leprae-specific antigen, Phenolic Glycolipid I (PGL-I) is now first defined in line 43. 

Introduction

The introduction overall fairy tries to highlight what the problem is, what the gaps are and what was done about it. However: (Line 118) The Dr. Marcelo Candia Reference Unit in Sanitary Dermatology (UREMC), I don’t understand the relevance of this clinic. Does it offer the standard testing; is it the Center of Excellence in Para? More clarity maybe needed for the reader.

Thank you for your suggestion. The Dr. Marcelo Candia Reference Unit in Sanitary Dermatology (UREMC) is the Pará state reference laboratory for the diagnosis and treatment of Hansen's disease and offers specialized multidisciplinary clinical care to its users. UREMC was responsible for 19.9% (489/2,548) of the total leprosy notifications in Pará in 2019. People with suspected signs of leprosy from all over the state are referred to UREMC, some who have been misdiagnosed for years. Several leprosy dermatologists, including Dr. Salgado, have been examining leprosy patients and diagnosing them based on clinical signs and symptoms for over 21 years. Follow-up laboratory tests can include bacilloscopy of skin lesions to determine the BI and/or presence of acid fast bacilli as well as determining the anti-PGL-I titer, both which can be confirmatory in the diagnosis and are used to determine the length of the treatment regimen, MDT-PB or MDT-MB. We have modified the text to reflect the importance of this leprosy reference laboratory in the care and treatment of leprosy patients. 

Line 144-Line 149 Are actually Methods and Materials in the introduction section. The study design has been highlighted as well as the study description. I think this should be moved to the methods section.

According to the STROBE checklist for cross-sectional studies (von Elm et al. Lancet, 2007), in the Introduction item #2 for Background/rationale states “Explain the scientific background and rationale for the investigation being reported.” These sentences are critically important to establish the rationale for investigating these two particular biomarkers in leprosy patients, healthy household contacts and healthy endemic control subjects that we examined. 

Line 149-line 153 this is actually the discussion section in the introduction. This part should be moved to the discussion section.

Again, according to the STROBE checklist for cross-sectional studies, in the Introduction item #3 for Objectives states “State specific objectives, including any prespecified hypothesis”. These sentences are critically important so that the specific objectives are made clear and a hypothesis about the potential risks for individuals who are double positive for both biomarkers is also stated clearly. 

Methods

Overall, the methods are poorly presented and hence the study cannot be reproduced. We don’t know what study design was used. A prospective study design was mooted in the introduction but it is not anywhere else. I advise the authors to look up the STROBE guidelines by Von Elm et al.

This the study is about probable diagnostic tests, the authors should tell us if the tests were done in series or in parallel since the results change depending on what is done.

There are some result tables within the methods section.

Thank you for your suggestion. According to the STROBE checklist for cross-sectional in series studies, we have included the following items in the Materials and Methods section and made some selective modifications in the text on the Study Design, Setting, Participants, Variables, Data measurements, Study size, Quantitative variables and Statistical methods. We have now added a new paragraph “Sampling design and methods”, lines 180-202, to further clarify how patients and household contacts were identified and sampled. As far as reproducibility, the ELISA anti-PGL-I assay and RLEP PCR described have been in use by us for over 12 years and both of these tests have been used by leprosy investigators all over the world, they are well-established and easily reproducible in well equipped laboratories. Based on item #10 for Study size the STROBE checklist states “Explain how the study size was arrived at”, Table 1 shows the number of individuals in each group (new leprosy patients, treated leprosy patients, healthy household contacts and healthy endemic controls) and from what cities they came from to answer this question. We have moved Table 1 to the Results section. 

Results

I still advise the authors to have a look at the STROBE guidelines. However, it is important to add the socio-demographics of the study subjects unless this is a secondary analysis. The socio-demographics will help us with generalization and also perspective.

The author should use the statistical methods mentioned in the results section to find out if they are significant. Currently all we have are the proportions. We are not sure if they are purely by chance.

Thank you for your suggestion. Sociodemographic aspects are extremely relevant for understanding the epidemiology of leprosy, historically recognized as a disease of poverty1 and sociodemographic data have previously been explored by other groups for meta-analysis studies2. Although this data was not the primary focus of this study, we did collect it from all of the study participants. In previous studies by us, we highlighted potential risk factors that likely lead to higher rates of leprosy in the Amazon region including living in a very high or hyperendemic area for leprosy, poverty, lack of clean water and sanitation in the house, high household density with more than 2 people sleeping per bedroom, poor nutritional status and lack of health care availability.3,4 Certainly the individuals in the leprosy patient, treated patient and healthy household contact groups are more disadvantaged and likely have multiple risk factors among the above, while those in the healthy endemic group who are mostly university students are better off and live in higher socioeconomic areas of the capital of Belem. We have added Table 3 (see below) that includes this sociodemographic information in the Results and a paragraph in the Discussion.

1. Nery JS, Ramond A, Pescarini JM, Alves A, Strina A, Ichihara MY, Penna MLF, Smeeth L, Rodrigues LC, Barreto ML, Brickley EB, Penna GO. 2019. Socioeconomic determinants of leprosy new case detection in the 100 Million Brazilian Cohort: a population-based linkage study. Lancet Glob Health 7: e1226-1236.

2. Pescarini JM, Strina A, Nery JS, Skalinski LM, Andrade KVF, Penna MLF, et al. 2018. Socioeconomic risk markers of leprosy in high-burden countries: A systematic review and meta-analysis. PLoS Negl Trop Dis 12(7): e0006622. doi.org/10.1371/journal.pntd.0006622

3. Barreto, J. G., D. Bisanzio, L.deS. Guimarães, J. S. Spencer, G. M. Vazquez-Prokopec, U. Kitron, and C. G. Salgado. 2014. Spatial analysis spotlighting early childhood leprosy transmission in a hyperendemic municipality of the Brazilian Amazon region. PloS Negl. Trop. Dis. 8(2):e2665. doi: 10.1371/journal.pntd.0002665.

4. Barreto JG, Bisanzio D, Frade MAC, Moraes TMP, Gobbo AR, Guimarães L de S, da Silva MB, Vazquez-Prokopec GM, Spencer JS, Kitron U, Salgado CG. 2015. Spatial epidemiology and serologic cohorts increase the early detection of leprosy. BMC Inf Dis 15: 527 doi 10.1186/s12879-015-1254-8.

As for the application of statistical tests: 

Lines 257-258: The statistical differences between groups were calculated by Student’s t-test 

Lines 280-282: When individuals in each group were divided into RLEP positive or negative and examined for anti-PGL-I titer, there were no statistical differences between the median O.D. values for RLEP positive versus negative individuals within the patient or HHC groups (Figure 2B).

Discussion

Overall the discussion is about the bio-markers and leprosy in general. Very little effort is made to discuss the results. Their significance and the recommendations if any, the overall recommendation is hinted all over the paper with no possible justification given from the results.

We have expanded the discussion of the results.

Reviewer #4: 

1) This is a good study trying to address the issue of transmission in leprosy.

Thank you for this comment. We feel that there are many unknowns about leprosy transmission that unfortunately cannot be answered because M. leprae is not cultivatable. 

2) In fact this investigation is trying to look at possible sources of infection in the community and their role in transmission of the disease.

Thank you for this comment. In hyperendemic areas as in all of the cities that we visited in this state, the percentage of people diagnosed by an experienced leprosy dermatologist based on clinical signs and symptoms ranged from 3.4% to 22.3%. These are truly staggering numbers, hundreds of time higher than the nationally reported new case detection rate (NCDR) reported yearly by the Brazil Ministry of Health database (SINAN), which usually relies on passive detection rates (currently around 1.3 new cases per 10,000 population). We have published these results in numerous articles over the years highlighting that active surveillance by us in cities in the state of Pará generally shows a new case detection rate of around 4% in children and 8% in household contacts in this region.

3) In Table 2 for all parameters the author should give number of MB and PB cases in each group

We have revised Table 2 that shows the breakdown of MB and PB cases for the new case and treated groups for each test combination (double positive, single positive and double negative). MB and PB case categories do not apply for the HHC and HEC groups.

 PGL+/ RLEP+ PB MB PGL-/RLEP+ PB MB PGL+/RLEP- PB MB PGL-/RLEP- PB MB

 n % n % n % n % n % n % n % n % n % n % n % n %

New cases (n = 87) 40 46.0 4 10 36 90 33 37.9 7 21.2 26 78.8 8 9.2 2 25 6 75 6 6.9 3 50 3 50

Treated

(n = 52) 12 23.1 4 33.3 8 66.7 11 21.2 4 36.4 7 63.6 14 26.9 2 14.3 12 85.7 15 28.8 5 33.3 10 66.7

HHC

(n = 296) 46 15.5 35 11.8 107 36.1 108 36.5 

HEC

(n = 31) 0 0% 0 0% 7 22.6 24 77.4 

Table 3. Demographic information of the four groups studied (new leprosy cases, treated patients, healthy household contacts and healthy endemic controls. Highlights of findings: There are overall slightly more females than males except in the treated group, but the ratio does not seem to be overly skewed. Some observations: it is apparent that all of the HEC have a clean source of drinking water (2 filtered, 29 bottled mineral water) while less than half of the people in the other three groups have access to this, possibly leading to more exposure to water borne pathogens such as amoeba and bacteria causing diarrheal disease. Income for HEC is also skewed to 100% having two minimum salaries or more whereas more than half of people in the other three groups have one minimum salary or less, suggesting conditions of poverty. Education levels for HEC are all at university level, while the other three groups have a majority with no education or only primary level schooling. Food deprivation is between 20% (one in 5) to 34% (one in three) for three groups, while HEC do not experience this problem of hunger at all. The number of people per household is around 3 for HEC while at least 5 for the other three groups, indicating overcrowded conditions. 

 Number of people per house Type of water used

for drinking/cooking Salary Education (highest grade) Receive governmental support

(%) Median age (range) Sex

(ratio M:F) Food deprivation

(%) Living in urban area

(%)

New cases

(n = 87) 5.5 Not treated: 4

Strained water: 39

Chlorinated: 2

Filtered: 36

Mineral water: 6 ≤ 1 minimum salary: 55

Up to two minimum salary: 21

Up to three minimum salary: 5

Greater than 3MS: 6 No Education: 66

Elementary School: 11

High school: 9

University education: 1 60/87

(69%) 25

(5-81) Female: 52

Male: 35

(0.6 : 1) 18/87

(20.7%) 81/87

(93.1)

Treated

(n = 52) 5 Not treated: 3

Strained water: 21

Chlorinated: 7 

Filtered: 12

Mineral water: 5 ≤ 1 minimum salary: 30

Up to two minimum salary: 13

Up to three minimum salary: 4

Greater than 3MS: 3 No Education: 30

Elementary School: 8

High school: 8

University education: 6 35/52

(67.3%) 45

(12-87) Female: 22

Male: 30

(1 : 0.58) 18/52

(34.6%) 42/52

(80.8%)

HHC

(n = 296) 5 Not treated: 15

Strained water: 147

Chlorinated: 32

Filtered: 70

Mineral water: 32 ≤ 1 minimum salary: 172

Up to two minimum salary: 81

Up to three minimum salary: 27

Greater than 3MS: 16 No Education: 106

Elementary School: 127 

High school: 52

University education: 11 193/296

(65.2%) 31

(6-79) Female: 160

Male: 136

(0.46 : 1) 102/296

(34.4%) 241/296

(81.4%)

HEC

(n = 31) 3 Not treated: 0

Strained water: 0 

Chlorinated: 

Filtered: 2

Mineral water: 29 ≤ 1 minimum salary: 0

Up to two minimum salary: 1

Up to three minimum salary: 2

Greater than 3MS: 28 No Education: 0

Elementary School: 0 

High school: 0

University education: 31 0/31

(0%) 33

(19-62) 

Female: 20

Male: 11

(0.35 : 1)

 0/31

(0%) 31/31

(100%)

4) In HHC it will be interesting to know the type of Index cases whether they are PB or MB

This data was not collected.

5) If skin smear BI available in Index case was there a correlation between high BI positivity and HHC positive

Since most of the work was done in the field, the BI of index cases was not assessed. However, we always do a bacilloscopy on leprosy cases attended at the UREMC reference center, resulting in a high correlation between our clinical definition and the BI.

6) Is there a correlation between smear positivity (If available) and positivity in the two tests performed?

We think there is since double positivity is highest in the new case group (46%) while it is markedly reduced in the treated patient group (23.1%) showing the treatment efficacy of MDT. 

7) In HHC was there any other household which had many members who were positive for any of the two tests and the author should mention about number of families which were positive. 

The size of each household visited varied considerably, from 3 to 12 and sampling at each house relied on who was at home at the time we visited with some household members away or at work. We cited one family as an example where out of twelve individuals with one index case, six other blood related individuals in this household were diagnosed that day and double positivity in this particular family was 75%. Although this was an extreme finding, there were other families where more than one person was diagnosed in the household that showed high rates of being double positive.

8) In HHC compare the positivity in different age groups and was there any correlation with higher age group?

We now have a new table that shows some of the variables for each of the groups (age, sex, income, household density, availability of clean water, etc.) (Table 3). 

9) The author needs to mention about the age groups of subjects and among the cases positive for any test in all groups, was there any child case?

The total population evaluated is made up of 466 individuals, of which 92/466 (19.7%) were under 15 yo. Of the 87 new cases, 38/87 (43.7%) were in children under 15 years old. We have added this to the Results section. 

10) There is no legend for the Tables mentioned

All tables have a legend that appears just above the tables.

Lines 237-240: Table 1. Operational classification and number of subjects per group and municipality. Characteristics of newly diagnosed leprosy patients, treated leprosy patients, healthy household contacts (HHC) and healthy endemic controls (HEC) from the seven cities surveyed.

Lines 294-297: Table 2. Correlation of RLEP and anti-PGL-I titer within each group. Double positive (PGL-I+/RLEP+), single positive (PGL-I+/RLEP- or PGL-I-/RLEP+) and double negative (PGL-I-/RLEP-) were calculated for each of the four groups.

11) How many HHC who were doubly positive were followed up if followed up and what was the outcome of that follow up?

We intend to do this in the future as we are getting funding for a project for long-term follow-up of contacts who were clinically healthy, but with positive laboratory results (ELISA and/or PCR). The main limiting factor is the lack of resources available for follow-up visits, which is certainly desirable in studies involving neglected chronic diseases like leprosy. Resources for continued leprosy surveillance projects are needed especially in hyperendemic areas, such as the state of Pará, with a territorial area of 1,248,000 km², and the municipalities selected in this project represent each one of the state's macro-regions. Follow-up visits to several of these cities are not possible logistically. For example, Senador José Porfirio is 516 miles away from the capital, Belém, and can only be reached by driving over 18 hours on the Trans-Amazon Highway. Another city, Breves, can only be reached by taking an overnight boat trip by river. In some cities visited we have provided the diagnosis and positivity data to the local health municipality leprosy control coordinators to perform follow-up with these families. Cities that are more easily visited like Acará, Belém and the surrounding metropolitan area (Mosqueiro) and others will be followed up in a future study. We have already published one study and a 2-year follow-up in the city of Castanhal where we found that in a follow-up of 10 individuals who came from anti-PGL-I positive households there would be a 90% chance of diagnosing one new case within 2 years and that individuals in anti-PGL-I positive households had a 2.7-fold higher risk of progressing to disease than those from negative households.1,2

1. Barreto JG, Bisanzio D, Guimarães L de S, Spencer JS, Vazquez-Prokopec GM, Kitron U, Salgado CG. 2014. Spatial analysis spotlighting early childhood leprosy transmission in a hyperendemic municipality of the Brazilian Amazon region. PLoS Negl Trop Dis 8(2): e2665. doi: 10.1371/journal.pntd.0002665.

2. Barreto JG, Bisanzio D, Frade MAC, Moraes TMP, Gobbo AR, Guimarães L de S, da Silva MB, Vazquez-Prokopec GM, Spencer JS, Kitron U, Salgado CG. 2015. Spatial epidemiology and serologic cohorts increase the early detection of leprosy. BMC Inf Dis 15: 527 doi 10.1186/s12879-015-1254-8.

Reviewer #5:

Leprosy is still a public health problem in some countries. In Brazil, the highest prevalence of the disease is observed in North, Northeast and Midwest regions. One important fact regarding the elimination of leprosy is the absence of a gold standard test for diagnosis that results in a significant number of hidden cases. Prophylactic treatment of contacts from multibacillary patients has been evaluated as an effective strategy to control the disease, but the identification of contacts with subclinical infection is important to determine the targets of chemoprophylactic strategy. Here, Silva and colleagues described that contacts that are positive for both RLEP and PGL-1 have latent disease and are at highest risk of progressing to clinical disease. This paper is interesting. However, I have some concerns:

1. Although the experimental design seems correct, authors need to describe the methodology for analyzing the data. How did they calculate sensitivity, specificity and accuracy? Please include in the methodology section.

Thank you for this question. Accuracy is the test ability to identify correctly subjects with leprosy (true positive) and healthy endemic controls (true negative). Sensitivity is the test ability to identify correctly subjects with leprosy (true positive). Specificity is the test ability to identify correctly subjects without leprosy, named as healthy endemic controls (true negative). All concepts described above were based on well-established terms of statistics and contextualized to leprosy disease1,2. Sensitivity, accuracy, and specificity were determined by the exact method of Clopper and Pearson3 using the GraphPad prism 6 (GraphPad Software, California, USA). 

1. Gurung P, Gomes CM, Vernal S, Leeflang MMG. 2019. Diagnostic accuracy of tests for leprosy: a systematic review and meta-analysis. Clin Microbiol Infect 25: 1315-1327.

2. Vogels CBF, Brito AF, Wyllie AL et al. 2020. Analytical sensitivity and efficiency comparisons of SARS-CoV-2 RT-qPCR primer-probe sets. Nat Microbiol doi.org/10.1038/s41564-020-0761-6

3. Clopper C and Pearson ES. 1934. The use of confidence or fiducial limits illustrated in the case of the binomial. Biometrika 26: 404-413. doi:10.1093/biomet/26.4.404

2. The "Discussion" is rather long and sometimes confusing. It contains common information that is not directly related to the specific topic of this study.

We will try our best to limit the Discussion directly to our results although we were asked to include additional data in Table 3 with discussion.

3. Several studies have reported that PGL-I is a marker of exposure, but not necessarily of infection. It is not clear in the discussion what is the hypothesis for the different profiles observed in the contacts. For example, PGL-I+/RLEP+ are the latent patients. But what is the hypothesis for PGL-I-/RLEP+? Please discuss it.

These are excellent questions. We would argue that in order to induce a positive anti-PGL-I response there must be an active infection. There is consensus and support in the literature that “the presence of circulating anti-PGL-I among healthy contacts was considered to indicate a subclinical infection”.1 However, not everyone who is infected has a positive antibody response, e.g. only around 20-40% of diagnosed PB patients have a positive titer because their bacillary load is low. Studies indicate that having a positive anti-PGL-I titer increases the risk of succumbing to leprosy by around 6-fold.2 Our hypothesis is that those individuals who are PGL-I-/RLEP+ are also subclinical, they just do not have a sufficient bacillary load to induce an anti-PGL-I response. We are attempting to use quantitative RT-PCR to correlate the bacillary load in earlobe SSS with the anti-PGL-I response. We have added a paragraph into the Discussion to clarify this:

The four different possible combinations of ELISA/PCR results can be cautiously interpreted in several ways. We propose that those individuals without clinical signs and symptoms of leprosy who are PGL-I+/RLEP+ have latent leprosy infection, allowing permissive growth to allow infection of M. leprae in the earlobe and spread to other sites in the skin and an antibody response. These individuals most resemble newly diagnosed patients, the majority of whom are double positive, and thus are at the greatest risk of progressing to disease. Individuals who are PGL-I+/RLEP- are infected but their functional cell mediated immune response has limited bacterial infection in the earlobe, which can evolve to a cure or can progress to paucibacillary disease. PGL-I-/RLEP+ individuals are also infected but the bacillary load has not increased to the point that induces an anti-PGL-I response. These individuals could either control the bacilli or progress to disease if the cell mediated response allows permissive growth and spread. Individuals who are double negative, PGL-I-/RLEP-, may not have been exposed to enough of a bacterial load to infect them or were more resistant to infection. These results could change over time depending on continued exposure to an untreated index case or other factors influencing a robust cell mediated immune response (co-infections, poor nutritional status).

1. Araújo S, Lobato J, Reis E de M, Souza DOB, Goncalves MA, Costa AV, Goulart LR, Goulart IMB. 2012. Unveiling healthy carriers and subclinical infections among household contacts of leprosy patients who play potential roles in the disease chain of transmission. Mem Inst Oswaldo Cruz 107 (Suppl I): 55-59.

2. Goulart IMB, Souza DOB, Marques CR, Pimenta VL, Goncalves MA, Goulart LR. 2008. Risk and protective factors for leprosy development determined by epridemiological surveillance of household contacts. Clin Vacc Immunol 15: 101-105.

---

## [Decision Letter · Decision Letter 1]

22 Jan 2021

PONE-D-20-19814R1

Latent leprosy infection identified by dual RLEP and anti-PGL-I positivity: Implications for new control strategies

PLOS ONE

Dear Dr. John S. Spencer, Greetings.

Thank you for submitting your revised manuscript "Latent leprosy infection identified by dual RLEP and anti-PGL-I positivity: Implications for new control strategies" to the PLOS ONE for consideration. This is an interesting manuscript in an important area, so the editors sought external views to add to our own. However, the reviewers' comments and recommendations were mixed. The primary concerns are regarding the presentation and feeble language. The quality of the English used throughout your manuscript does not currently meet our minimum requirements, as there are substantial incorrect sentence constructions and grammatical errors throughout, obscuring the message the authors want to convey. After discussing the paper further, the editors felt that the manuscript need significant changes, mostly the sentences rephrasing. We recommend consulting native speakers.

By considering the lengthy review process and considerable improvement in the revised manuscript, we would like to give the authors a final chance to fix the issues. If you can address the points raised by editors and reviewers, we would encourage you to submit a revised manuscript. Once we have received your revised manuscript, a decision will be made, which we expect by 22 February 2021.

 If you will need more time than this to complete your revisions, please reply to this message or contact the journal office at plosone@plos.org. Please include the following items when submitting your revised manuscript:

We look forward to receiving your revised manuscript.

Kind regards,

**Dr. Supram Hosuru Subramanya, Ph.D.**

**Academic Editor**

PLOS ONE

**Additional Editor Comments:**

The quality of the English used throughout your manuscript is not acceptable for publication. Many sentences are of 4 to 5 lines; sometimes, it's impossible to understand what authors want to convey. I suggest to break the long sentence and make it concise and clear. The first sentence of the abstract itself confusing. I feel like reading google translated texts. Also, please check the word "Positivity" in the title is suitable over there????.

Reviewers' comments:

Reviewer's Responses to Questions

**Comments to the Author**

1. If the authors have adequately addressed your comments raised in a previous round of review and you feel that this manuscript is now acceptable for publication, you may indicate that here to bypass the “Comments to the Author” section, enter your conflict of interest statement in the “Confidential to Editor” section, and submit your "Accept" recommendation.

Reviewer #1: All comments have been addressed

Reviewer #3: All comments have been addressed

Reviewer #4: All comments have been addressed

Reviewer #5: All comments have been addressed

2. Is the manuscript technically sound, and do the data support the conclusions?

Reviewer #1: Yes

Reviewer #3: No

Reviewer #4: Yes

Reviewer #5: Yes

3. Has the statistical analysis been performed appropriately and rigorously? 

Reviewer #1: Yes

Reviewer #3: No

Reviewer #4: Yes

Reviewer #5: Yes

4. Have the authors made all data underlying the findings in their manuscript fully available?

Reviewer #1: Yes

Reviewer #3: No

Reviewer #4: Yes

Reviewer #5: Yes

5. Is the manuscript presented in an intelligible fashion and written in standard English?

Reviewer #1: Yes

Reviewer #3: No

Reviewer #4: Yes

Reviewer #5: Yes

6. Review Comments to the Author

Reviewer #1: (No Response)

Reviewer #3: Overall, PONE-D-20-19814R1 is better written compared to PONE-D-20-1981. Some of my comments were addressed in the review however; I still have a few more comments.

Introduction

The introduction does clearly explain the problem, but it could be summarized. The flow is good till line 109, however after that what was done should have been summarized.

I understand the importance of UREMC in confirmation of Leprosy disease however, it is just a referral site for the region and the authors should state likewise. However, if the authors must include it in the paper, then they should shift it to the methods section, and then give details about it and what role it plays including the staff who they are and what they do.

Line 145-to Line 154 is actually Methods and Materials and the results in the introduction section. Despite the authors attempt to highlight that they are giving a rationale for the investigation. They actually stated the study design and what they found. If they were tempting to give the hypothesis, there is need for re-wording

“We show that the majority of newly 151 diagnosed leprosy cases with clinical symptoms are positive for both of these 152 biomarkers of infection suggesting the possibility that household contacts who are also 153 double positive may have latent disease and should be carefully monitored through 154 follow-up examinations.”

Methods

The methods and materials have improved however the study cannot be reproduced.

It may be important to state the sampling methods used to get the evaluation areas. The authors do not that it was divided into Rural and Urban. But there is no other mention of any sampling technique and why it was chosen.

It may be important to find out how the authors arrived at 466 individuals with 87 newly infected, 52 former patients, 296 household contacts and 31 healthy endemic controls. Was this also random or it was the available data?

There is completely no mention of how the data was collected or stored prior to being analyzed

Results

Line 255 to 256, in the results section we expect to get the results got from the use of the t-test not what statistical test was used.

Discussion

This may seem like a contradiction on my last review, but I think there is need summarize the key finding of the study and discuss only those..

Reviewer #4: 1) Line 268 _Within the 268 new case group, detection of RLEP was 87.3% (62/71) in MB cases and 68.8% (11/16) in PB cases. When cases were subdivided according to Ridley-Jopling classification for the different forms across the disease spectrum, RLEP amplification was positive in 57.1% (3/7) for the indeterminate form, 80% (4/5) for TT, 87.5% (42/48) for BT, 84.2% (16/19) for BB and 100% for BL and LL (4/4). In addition, four cases of primary neural form (PNL) were diagnosed, and all were positive (100%, 4/4).

Similar results can be documented for anti-PGL-I.

2.) comment no 9) The author needs to mention about the age groups of subjects and among the cases positive for any test in all groups, was there any child case?

The total population evaluated is made up of 466 individuals, of which 92/466 (19.7%) were under 15 yo. Of the 87 new cases, 38/87 (43.7%) were in children under 15 years old. We have added this to the Results section

However, we did not find the above results mentioned in the revised manuscript. Also, findings on the number of children in new, treated and HHC group, will give information about the transmission.

3) Line 337 - For this reason, we have been using

338 less invasive methods, namely taking samples of blood and earlobe SSS.

(In response to our comment no 5) If skin smear BI available in Index case was there a correlation between high BI positivity and HHC positive

Since most of the work was done in the field, the BI of index cases was not assessed.

However, we always do a bacilloscopy on leprosy cases attended at the UREMC reference center, resulting in a high correlation between our clinical definition and the BI. )

We suggest that to obtain the BI, SSS from all earlobe, forehead and active lesion can be taken on slide, heat fixed, stained by ZNCF and quantified under oil immersion. There is no need of taking invasive punch biopsy.

Reviewer #5: Authors have adequately addressed my comments raised in a previous round of review and the manuscript is now acceptable for publication.

7. PLOS authors have the option to publish the peer review history of their article (what does this mean?). If published, this will include your full peer review and any attached files.

Reviewer #1: **Yes: **Azin Ayatollahi, MD

Reviewer #3: **Yes: **Michael Kakinda

Reviewer #4: No

Reviewer #5: **Yes: **Roberta Olmo Pinheiro

---

## [Author Response · Author response to Decision Letter 1]

25 Feb 2021

The file "Response to Reviewers" has been attached addressing all questions.

---

## [Decision Letter · Decision Letter 2]

19 Mar 2021

PONE-D-20-19814R2

Latent leprosy infection identified by dual RLEP and anti-PGL-I positivity: Implications for new control strategies

PLOS ONE

Dear Dr. John S. Spencer,

Thank you for submitting your revised manuscript " Latent leprosy infection identified by dual RLEP and anti-PGL-I positivity: Implications for new control strategies" to PLOS ONE. Peer review of your manuscript is now completed. Based on these reports and my own assessment as Editor, I am pleased to inform you that it is potentially acceptable for publication in PLOS ONE once you have carried out some essential revisions suggested by a reviewer. Currently, reviewer-3 has raised few genuine concerns which need to be addressed. If you can't address all the issues, I recommend discussing those points in the discussion part of the manuscript or state as a limitation of the study.

Some manuscripts require many rounds of revisions, so this is a standard but necessary stage of the editorial process. Therefore, I invite you to revise your paper, considering the points raised during the review process.

Please go over your manuscript text and ensure that it is written concisely and clearly. At the same time, we ask you to make sure your manuscript complies with our format requirements detailed on the journal website.

We look forward to receiving your revised manuscript.

Kind regards,

**Dr. Supram Hosuru Subramanya, Ph.D.**

**Academic Editor**

**PLOS ONE**

Journal Requirements:

Reviewers' comments:

Reviewer's Responses to Questions

**Comments to the Author**

1. If the authors have adequately addressed your comments raised in a previous round of review and you feel that this manuscript is now acceptable for publication, you may indicate that here to bypass the “Comments to the Author” section, enter your conflict of interest statement in the “Confidential to Editor” section, and submit your "Accept" recommendation.

Reviewer #1: All comments have been addressed

Reviewer #3: All comments have been addressed

Reviewer #4: All comments have been addressed

Reviewer #5: All comments have been addressed

2. Is the manuscript technically sound, and do the data support the conclusions?

Reviewer #1: Yes

Reviewer #3: No

Reviewer #4: Yes

Reviewer #5: Yes

3. Has the statistical analysis been performed appropriately and rigorously? 

Reviewer #1: Yes

Reviewer #3: No

Reviewer #4: Yes

Reviewer #5: Yes

4. Have the authors made all data underlying the findings in their manuscript fully available?

Reviewer #1: Yes

Reviewer #3: No

Reviewer #4: Yes

Reviewer #5: Yes

5. Is the manuscript presented in an intelligible fashion and written in standard English?

Reviewer #1: Yes

Reviewer #3: Yes

Reviewer #4: Yes

Reviewer #5: Yes

6. Review Comments to the Author

Reviewer #1: (No Response)

Reviewer #3: Overall, PONE-D-20-19814R2 is better written compared to PONE-D-20-1981 R1. Some of my comments were addressed in the review however; I still have a few more comments.

Introduction

The introduction does clearly explain the problem, but it could be summarized. The flow is good till line 109, however after that what was done should have been summarized.

Line 145-to Line 154 is actually Methods and Materials and the results in the introduction section. Despite the authors attempt to highlight that they are giving a rationale for the investigation. They actually stated the study design and what they found. If they were tempting to give the hypothesis, there is need for re-wording

Methods

The methods and materials have improved however the study cannot be reproduced.

It may be important to state the sampling methods used to get the evaluation areas. The authors do not that it was divided into Rural and Urban. But there is no other mention of any sampling technique and why it was chosen.

It may be important to find out how the authors arrived at 466 individuals with 87 newly infected, 52 former patients, 296 household contacts and 31 healthy endemic controls. Was this also random or it was the available data?

There is completely no mention of how the data was collected or stored prior to being analyzed

Results

Line 255 to 256, in the results section we expect to get the results got from the use of the t-test not what statistical test was used.

Discussion

This may seem like a contradiction on my last review, but I think there is need summarize the key finding of the study and discuss only those.

Reviewer #4: It is observed that the comments made earlier in the review has been addressed satisfactorily. The comments made have been responded to and have been also incorporated in the manuscript. As the comments have been addressed and incorporated in the manuscript the issues related have been clear and it therefore helps to relate and convey the objectives and purpose of the study and the results arising out of the study which are very important and have epidemiological implications.

Reviewer #5: (No Response)

7. PLOS authors have the option to publish the peer review history of their article (what does this mean?). If published, this will include your full peer review and any attached files.

Reviewer #1: **Yes: **Azin Ayatollahi, MD

Reviewer #3: **Yes: **Michael Kakinda

Reviewer #4: No

Reviewer #5: **Yes: **Roberta Olmo Pinheiro

---

## [Author Response · Author response to Decision Letter 2]

1 Apr 2021

We have responded to all comments from Reviewer #3 in Response to Reviewers with notations in the Revised Manuscript with Track Changes as necessary.

---

## [Editor Report · Decision Letter 3]

19 Apr 2021

PONE-D-20-19814R3

Latent leprosy infection identified by dual RLEP and anti-PGL-I positivity: Implications for new control strategies

PLOS ONE

Dear Dr. John S. Spencer,

Thank you for submitting your revised Manuscript to PLOS ONE. It has been suggested to add the "Limitations of the Study" to the manuscript after the discussion part in the previous editorial decision. However, I could not see it in the revised submission. If you have added it, please highlight it in colored text, and if you are not willing to add it, please respond with reasons in the letter. Now we are returning your manuscript and invite you to submit a revised version that addresses the points raised during the editorial process..

We look forward to receiving your revised manuscript.

Kind regards,

**Dr. Supram Hosuru Subramanya, Ph.D.**

Academic Editor

PLOS ONE
---

## [Author Response · Author response to Decision Letter 3]

27 Apr 2021

We have added a paragraph "Limitations of the Study" after the Discussion as requested by the editor.

---

## [Editor Report · Decision Letter 4]

30 Apr 2021

Latent leprosy infection identified by dual RLEP and anti-PGL-I positivity: Implications for new control strategies

PONE-D-20-19814R4

Dear Dr. John S. Spencer,

We’re pleased to inform you that your manuscript has been judged scientifically suitable for publication and will be formally accepted for publication once it meets all outstanding technical requirements.

Kind regards,

Supram Hosuru Subramanya, Ph.D.

Academic Editor

PLOS ONE
---

## [Editor Report · Acceptance letter]

5 May 2021

PONE-D-20-19814R4 

Latent leprosy infection identified by dual RLEP and anti-PGL-I positivity: Implications for new control strategies. 

Dear Dr. Spencer:

I'm pleased to inform you that your manuscript has been deemed suitable for publication in PLOS ONE. Congratulations! Your manuscript is now with our production department. 

Kind regards, 

on behalf of

Dr. Supram Hosuru Subramanya 

Academic Editor

PLOS ONE